# Assessing Learning Preferences of Tertiary Education Students in Jordan Post COVID-19 Pandemic and the Associated Perceived Stress

Iman A. Basheti [1,2,*] , Thafer Yusif Assaraira [3], Nathir M. Obeidat [4], Fawwaz Al-abed Al-haq [5] and Mashhoor Refai [6]

1    Department of Clinical Pharmacy and Therapeutics, Applied Science Private University, Amman 11931, Jordan
2    Faculty of Pharmacy, The University of Sydney, Sydney 2006, Australia
3    English Department, Mutah University, Al-Karak 61710, Jordan
4    Department of Internal Medicine, The University of Jordan Amman, Amman 11931, Jordan
5    Department of English Language and Literature, Hashemite University Zarqa, Zarqa 13110, Jordan
6    King Abdullah II School of Engineering, Princess Sumaya University for Technology, Amman 11931, Jordan
*    Correspondence: dr_iman@asu.edu.jo

**Abstract: Background**: The COVID-19 pandemic was associated with extensive lockdown strategies which included universities, forcing educational administrations to implement online learning and acknowledging the countless consequences it would have on the educational process. Those prompt changes highlighted the importance of online learning effects on educational outcomes. **Aim**: To assess students' learning preferences and the stress associated with online and face-to-face learning. **Methods**: This is a multi-center cross-sectional study, employing a web-based Google Forms, which was conducted among four universities in Jordan. The survey assessed students' demographic characteristics, educational methods received, assessment of factors that may have influenced students' stress, and assessment of 'stress' using the Perceived Stress Scale (PSS). **Results**: Among 1241 participating students, most of the students preferred face-to-face learning (43.3%), although the majority believed that online learning is less stressful (42.2%). The majority believed that face-to-face learning is efficient (42.7%), and that online learning is moderately efficient (38.4%), while many (35.3%) reported that the future of learning will be blended 50/50 between online and face-to-face learning. The mean score of PSS was 20.88, with 62.9% reported to have experienced moderate perceived stress, and 22.4% experienced high perceived stress. **Conclusions**: Although Jordanian university students prefer face-to-face learning over online learning, they believe that online learning can be less stressful. In addition to that, Jordanian students experienced a high mean of the PSS score, with more than 20% of students reporting high perceived stress.

**Keywords:** education; face-to-face; online; COVID-19; stress; Jordan

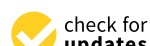



## 1. Introduction

Coronavirus disease 2019 (COVID-19) is a highly contagious viral illness caused by SARS-CoV-2, and during the last few years, has had catastrophic consequences on the world's population. The first COVID-19 case was reported in late December 2019 in Wuhan, China, after which SARS-CoV-2 spread rapidly across the globe. Such an alarming situation prompted the World Health Organization (WHO) to declare it a global pandemic on 11 March 2020 [1]. The COVID-19 symptoms varied from asymptomatic indications to mild, moderate, or severe symptomatic indications, with patients either experiencing self-resolution or requiring hospitalization [2].

Most countries, including Jordan, had restrictive policies implemented such as social distancing, banning of gatherings, and mobility restrictions, with many countries, also implementing lockdown strategies [3]. Such lockdown strategies had a positive impact

on decreasing COVID-19 cases. For instance, data proved that the outcome of lockdown strategies, which decreased the number of COVID-19-infected people, was achieved after ten days of implementing lockdown policies [4]. However, with such positive outcomes resulting from implementing such policies, undesirable outcomes started to emerge, including psychological effects in society [5]. Hence over time, it became imperative to vigilantly evaluate the benefits of mandatory mass quarantine against the possible odds of psychological distress in society for future cognizant decisions to be implemented. A recent textual analysis of 5780 publications carried out during this pandemic imposes the need for a global research alliance in order to address the knowledge gaps in a country to country based approach [6].

Tertiary education institutes (all formal post-secondary education, including public and private universities, colleges, technical training institutes, and vocational schools) were highly affected by the lockdown strategies [7]. Today, there are around 220 million tertiary education students in the world, most of whom were affected by the lockdowns imposed worldwide. To compensate for this, online learning became the most suitable method approved by authorities to allow the continuation of education. In addition, all ceremonies and activities were canceled, including graduation ceremonies, workshops, and conferences [8].

Online learning affected the educational process in many aspects, and discrepancies in consequences were seen based on the student's area of residence; in some residential areas, students were ready for the online learning experience and showed a positive attitude [9,10], while in other residential areas, more challenges were met due to a lower a socio-economic status for example, which played a significant role in students' compliance with the process [11–13]. In addition, the perception of students toward the online learning experience during the pandemic varied. Some students were overwhelmed with the new process, and the lack of availability of hardware equipment such as laptops and access to the internet which hindered achieving the needed outcomes. Others showed excellent adherence to online learning teaching, and unexpectedly showed higher preference over the earlier format of face-to-face education [14,15].

This study aimed to assess the perspective of tertiary education students living in Jordan on the effectiveness of online learning imposed since the start of the pandemic, their perspectives on the future of online learning, and the stress associated with online learning.

## 2. Methods

### 2.1. Study Design and Participants

This cross-sectional study was conducted via a survey presented to the potential participant via web-based Google Forms. The web-based Google Forms used is considered to be a reliable platform with regard to data protection as only the main researcher who has access to the collected data from the participants can enter the participants' responses. Moreover, this was clarified to the students prior to their participation through the informed consent paragraph added at the beginning of the survey. Data collection was conducted between 29 June and 12 September 2022. Inclusion criteria were participants living in Jordan and attending a public or private university. It was clarified at the start of the form that participation in the study was voluntary and did not pose any risk to the participants and an informed consent was obtained from the participants before moving forward to complete the questions in the survey.

### 2.2. Survey Development

After an extensive review of the literature, the survey's initial draft was constructed. Afterward, the research team developed the survey by formulating related items that would address the objectives of the current study. Five independent academics with prior research experience evaluated the survey's first draft to ensure face and content validity. They were requested to evaluate the items' reading comprehension, relevance, and clarity of words. As a final point in the survey development, the research team edited the survey's

items according to the evaluators' feedback. The survey was piloted with a small group of students ($n = 50$) in order to ensure its' comprehensibility and readability, as well as ensure the survey's applicability to Jordanian students.

The final version of the survey was organized into four main sections addressing the topics of interest in this study. The first section was designed to collect participants' demographic data including age, gender, living place, living condition, marital status, smoking status, chronic disease, medications taken if any, number of cups of coffee per day, having a family member diagnosed with anxiety and/or depression, and type of university the participant is studying at. The second section included items to assess the educational methods received by the participants and their perspectives on it (face-to-face, online, and blended education). The third section included factors that may have affected students' stress levels during the face-to-face and online learning they received during the past one year. Participants were requested to indicate their degree of agreement with each of the statements ($n = 18$) using a 5-point Likert scale (strongly agree, agree, neutral, disagree, and strongly disagree). The fourth section aimed to assess the participants' stress levels using the published and validated Perceived Stress Scale (PSS).

### 2.3. The Perceived Stress Scale (PSS)

The PSS is a predesigned, validated, and published instrument that is used to assess stress levels among participants, developed in 1983. The items in the PSS inquire about participants' feelings during the previous month, where participants are prompted to indicate how frequently they experienced each of the feelings they were asked about. The PSS is a 10-items scale, with 5 possible score options for each item (never, almost never, sometimes, fairly often, and very often).

### 2.4. Survey Implementation

Study participants were recruited mainly through social media (e.g., Facebook and WhatsApp) or by sending e-mails explaining the aim of the study. Those who were interested in participating were asked to click on a link to examine the study's ethical review board-approved information page before proceeding to the study survey. The survey was designed to take less than 10 minutes to be completed. Participants were given the option to complete the survey in English or Arabic.

### 2.5. Ethical Approval

Ethics approval for the study was obtained from the Faculty of Pharmacy, Applied Science Private University Ethics Committee (study approval number: 2022-PHA-15).

### 2.6. Sample Size

Using the Epi Info software, the sample size was calculated using a 95% confidence level, 50% expected frequency, 5% acceptable margin of error, and a design effect of 1.0. Three hundred and eighty-four participants were the minimum representative sample size needed for the study.

### 2.7. Statistical Analysis

Following data collection, the survey responses were coded and entered into a customized database using the Statistical Package for the Social Sciences (SPSS), Version 24.0 (IBM Corp., Armonk, New York, NY, USA). Frequency and percentages were used to present the qualitative variables, whereas continuous variables were presented as mean (standard deviation).

The linear regression was used to screen for certain variables affecting participants' stress management including age, gender, living place, living condition, marital status, smoking status, chronic disease, taking medication, cups of coffee per day, having a family member diagnosed with anxiety or depression, and type of university. First, simple linear regression was conducted; a variable that had a $p$-value of less than 0.25 was deemed

suitable to enter into the multiple linear regression model. For the multiple regression model, a variable that had a *p*-value equal to or less than 0.05 was considered to be statistically significant. All variables were chosen after confirming their independence, thus, a tolerance value of less than 0.20 and variance inflation factor of less than 5 were checked to ensure the absence of multicollinearity between the independent variables.

## 3. Results

The study's participants (*n* = 1241) had a mean age of 21.02 years (SD = 3.616), with the majority being females (60.2%). Regarding the living place, most of the participants (78.9%) were residing in Amman (the capital), and more than 90.0% (*n* = 1134) were living in urban areas. Being students, about 95.0% of the study's participants were single (*n* = 1178). The detailed demographic characteristics of the study's participants are shown in Table 1.

**Table 1.** Demographic characteristics of the study participants (*n* = 1241).

| Parameter | n (%) |
|---|---|
| Gender | |
| • Male | 494 (39.8) |
| • Female | 747 (60.2) |
| Living place | |
| • Amman (the capital) | 979 (78.9) |
| • Other cities | 262 (21.1) |
| Living condition | |
| • Rural areas | 107 (8.6) |
| • Urban areas | 1134 (91.4) |
| Marital Status | |
| • Married | 45 (3.6) |
| • Single | 1178 (94.9) |
| • Divorced | 9 (0.7) |
| • Widowed | 9 (0.7) |
| Having sleep issues | |
| • Always | 185 (14.9) |
| • Often | 350 (28.2) |
| • Sometimes | 353 (28.4) |
| • Rarely | 268 (21.6) |
| • Never | 85 (6.8) |
| Having stress management issues | |
| • Always | 165 (13.3) |
| • Often | 331 (26.7) |
| • Sometimes | 354 (28.5) |
| • Rarely | 286 (23.0) |
| • Never | 105 (8.5) |
| Having a family member diagnosed with anxiety and/or depression previously | |
| • Yes | 217 (17.5) |
| • No | 1024 (82.5) |

With regards to the quality of sleep that participants receive, about 43.1% of them (*n* = 535) reported that they always/often have sleeping issues. As for stress management, 40.0% (*n* = 496) of participants reported that they always/often have stress management issues. Out of the study participants (*n* = 1241), 217 (17.5%) documented that one of their family members was diagnosed with anxiety or depression previously (Table 1).

More than 70.0% of students were non-smokers (*n* = 902). With regards to participants' health conditions, 94.4% (*n* = 1172) did not have any chronic diseases, whereas about 10.0% (*n* = 121) were taking medications. With regards to caffeine consumption, 38.9% of the participants (*n* = 483) documented that they do not consume any coffee, 38.4% (*n* = 477) documented that they only consume up to two cups of coffee daily (their source of caffeine), 19.4% (*n* = 241) reported that they consume between two to four cups a day, and 3.4% (*n* = 40) documented that they consume more than four cups of coffee a day (Figure 1).

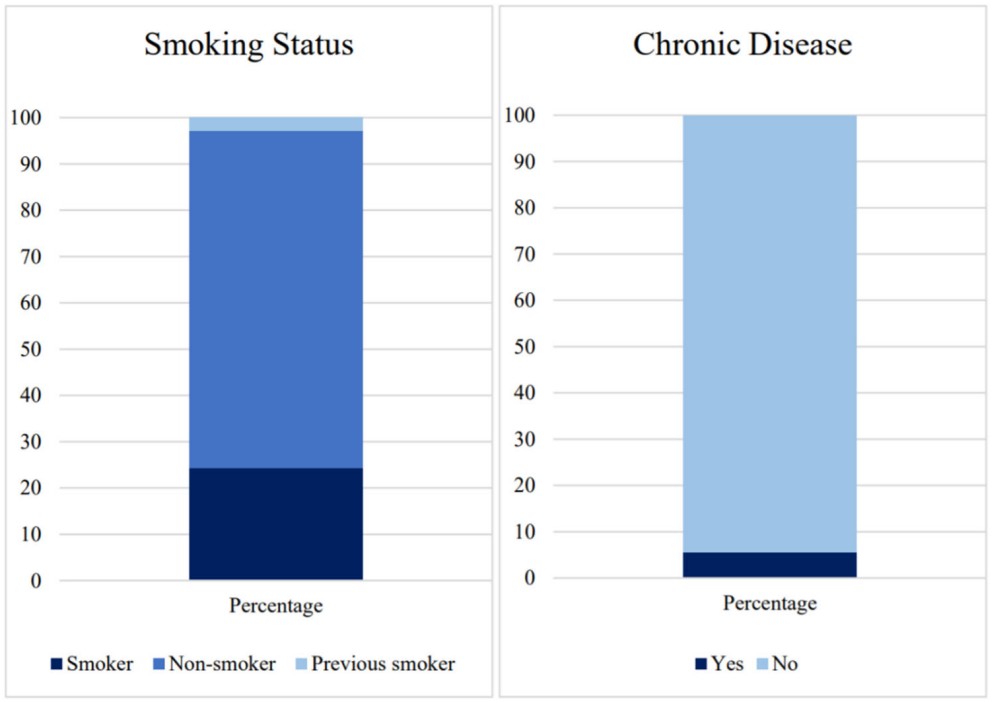

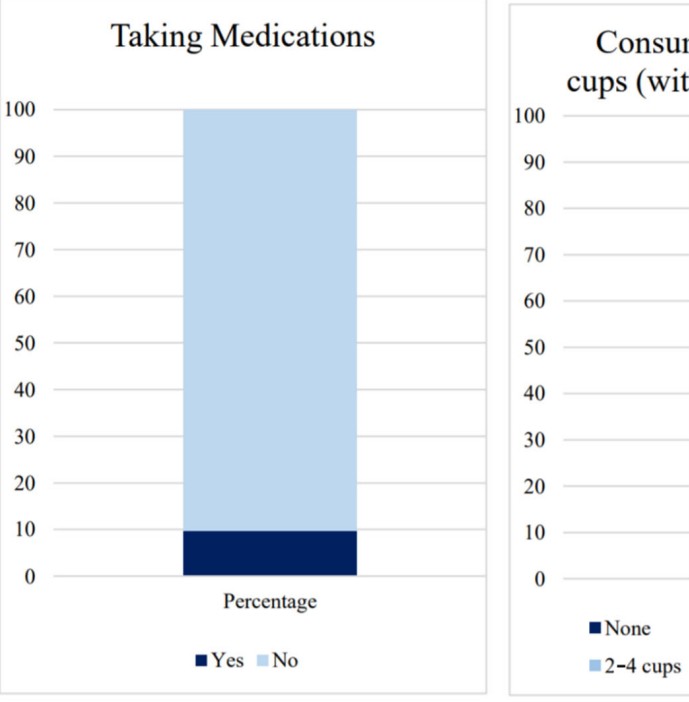

**Figure 1.** Lifestyle information among study participants (*n* = 1214).

Participants were mainly from Princess Sumaya University for Technology (PSUT; a private university founded in 1991 by the Royal Scientific Society to serve as its academic arm, located in the capital Amman; 40.6%,), Applied Science Private University (ASU; founded in 1991; located in the capital Amman; 22.4%), the Hashemite University (HU; a public university established in 1995 by a royal decree, located in the vicinity of the city of Zarqa; 20.0%), and the University of Jordan (UJ, a public university; located in the capital Amman; founded in 1962 by royal decree, being the largest and oldest institution of higher education in Jordan; 16.0%,)

The majority of the participants (96.0%) were undergraduate, with 31.5% majoring in a field of study related to medicine and health ($n = 391$), 18.9% majoring in information technology ($n = 235$), 14.7% majoring in engineering ($n = 182$), 8.1% majoring in humanities studies ($n = 101$), 5.4% majoring in a field related to science ($n = 67$), and 21.4% ($n = 265$) majoring in other specialties. About one-quarter of the participants ($n = 307$) were in their 3rd academic years (Table 2).

**Table 2.** University information of the study participants ($n =1241$).

| Parameter | n (%) |
|---|---|
| **University name** | |
| • The University of Jordan (UJ) | 198 (16.0) |
| • The Hashemite University (HU) | 248 (20.0) |
| • Applied Science Private University (ASU) | 278 (22.4) |
| • Princess Sumaya University for Technology (PSUT) | 504 (40.6) |
| • Others | 13 (1.0) |
| **Study level** | |
| • Diploma | 26 (2.1) |
| • Undergraduate (bachelor's degree) | 1191 (96.0) |
| • Postgraduate studies | 24 (1.9) |
| **Major at university** | |
| • Medical and health majors | 391 (31.5) |
| • Scientific majors | 67 (5.4) |
| • Engineering majors | 182 (14.7) |
| • Humanities majors | 101 (8.1) |
| • Information technology majors | 235 (18.9) |
| • Other | 265 (21.4) |
| **Academic year of students** | |
| • 1st year | 317 (25.5) |
| • 2nd year | 255 (20.5) |
| • 3rd year | 307 (24.7) |
| • 4th year | 281 (22.6) |
| • 5th year | 54 (4.4) |
| • 6th year | 27 (2.2) |

As shown in Figure 2, when participants were asked about their perception of the educational methods they received at their university, about 43.3% of them chose 'face-to-face' education as their preferred educational approach, 41.0% chose 'blended methods' (a mix between online learning and face-to-face education), and only 15.7% chose 'online education'. On the other hand, 'online education' was chosen the most as the educational method that caused less stress (42.2%).

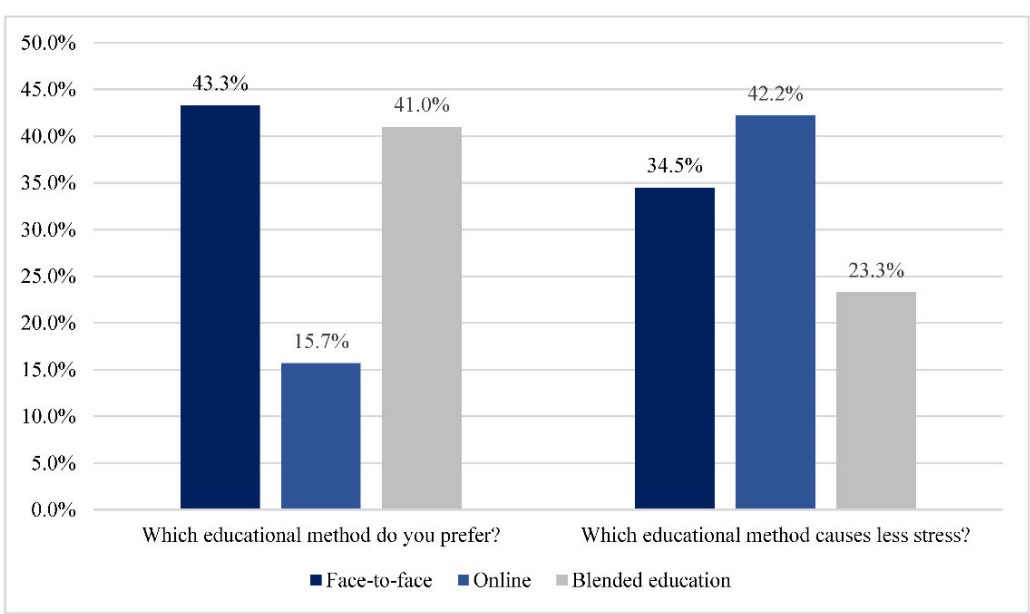

**Figure 2.** Participants' perception towards the educational methods applied at their universities (*n* = 1241).

Regarding the efficiency of students' learning in terms of the educational benefits (Figure 3), 34.6% of the participants reported that 'online learning' is efficient/highly efficient (*n* = 430), while 70.1% reported that 'face-to-face' learning is efficient/highly efficient (*n* = 870).

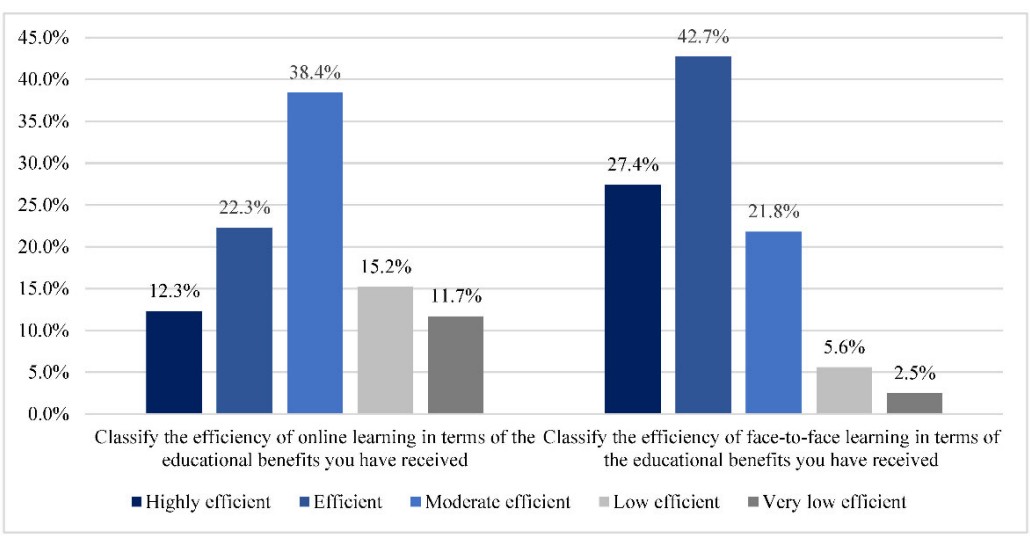

**Figure 3.** Participants' perception of the efficiency of educational methods applied at their university (*n* = 1241).

Forecasting the future, more than one-third (35.3%, *n* = 438) of participants believed that the education will be delivered '50% online and 50% face-to-face' after 10 years in Jordan (Figure 4). Many (28.0%; *n* = 347) chose '70% face to face and 30% online', while others (20.5%; *n* = 255) chose '30% face to face and 70% online'.

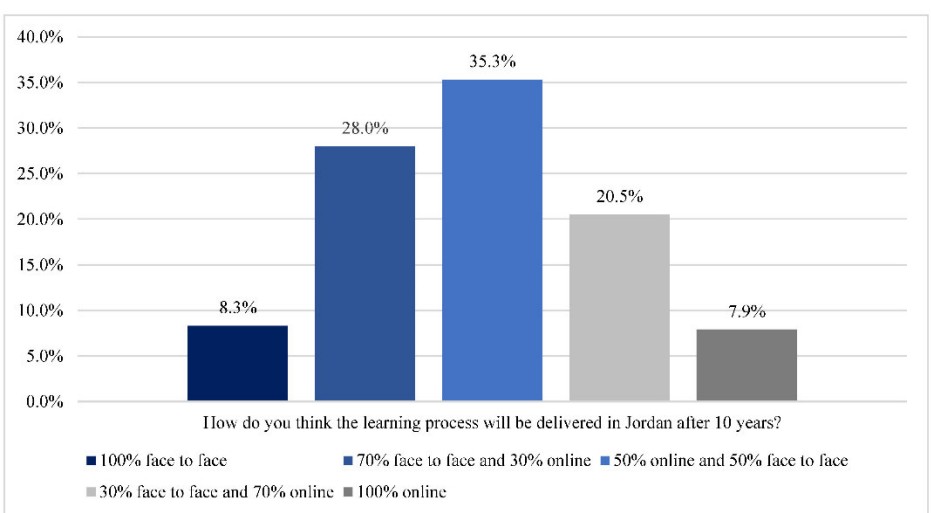

**Figure 4.** Participants' perception towards education in Jordan after 10 years (*n* = 1241).

The majority (63.0%) of participants agreed/strongly agreed that the learning process they forecasted for the future of education would reduce their stress levels, while 15.3% disagreed/strongly disagreed, and 21.7% were neutral, i.e., they could not judge.

During face-to-face education, the most factor that caused stress to the study participants was 'not being able to attend the lectures at a time that suits them'; 63.3% agreed/strongly agreed, followed by 'the higher study load during face-to-face learning'; 61.4% agreed/strongly agreed. On the other hand, the lowest reported factor that caused stress during face-to-face learning among the participants was 'meeting their peers (colleagues) at university'; 14.4% agreed/strongly agreed.

During online education, the most factor that caused stress to the study participants was 'using one-way questions during exams'; 84.6% agreed/strongly agreed, followed by 'short exam time increases my stress'; 73.4% agreed/strongly agreed. On the other hand, the lowest reported factor that caused stress during online learning among the participants was 'dealing with new platforms'; 36.1% agreed/strongly agreed. The mean percentage of agreed/strongly agreed on answers during face-to-face education was 41.27% while the mean percentage of agreed/strongly agreed on answers during the online education was 59.48%; this indicates that more students agreed/strongly agreed with factors that caused stress with the online learning than they did with factors that caused stress with the face-to-face education.

A list of all factors that caused stress during face-to-face and online education is shown in Table 3.

The mean total score of the PSS for all the study participants (*n* = 1241) was 20.88 (SD = 7.284; out of 40). Interpreting such scores shows that more than sixty percent of the participants (62.9%, *n* = 780) had moderate perceived stress (score = 14–26), whereas 22.4% (*n* = 278) had high perceived stress (score = 27–40), and 14.7% (*n* = 183) had low perceived stress (score = 0–13). The answer to each item is shown in Table 4.

Multiple linear regression analysis of factors affecting anger management highlighted that participants' age, gender, smoking status, medication intake, and caffeine consumption affected anger management, hence stress levels (Table 5). Thus, being a female (Beta = 0.289, *p*-value ≤ 0.001), taking medications (Beta = 0.273, *p*-value = 0.027), or drinking two cups of coffee or more with caffeine (Beta = 0.172, *p*-value = 0.029) is significantly associated with having anger management issues. On the other hand, being a non-smoker, or being younger is significantly associated with having anger management issues.

**Table 3.** Participants' responses regarding the factors that may increase their stress during face-to-face and online learning (*n* = 1241).

| Statement | Face-to-Face Learning | | | | |
|---|---|---|---|---|---|
| | Strongly Agree *n* (%) | Agree *n* (%) | Neutral *n* (%) | Disagree *n* (%) | Strongly Disagree *n* (%) |
| Coming to university everyday increases my stress | 146 (11.8) | 288 (23.2) | 206 (16.6) | 278 (22.4) | 323 (26.0) |
| Meeting my peers (colleagues) at university increases my stress | 77 (6.2) | 99 (8.0) | 181 (14.6) | 471 (38.0) | 413 (33.3) |
| Getting out of home (my comfort zone) increases my stress | 106 (8.5) | 176 (14.2) | 246 (19.8) | 406 (32.7) | 307 (24.8) |
| Having to move between lectures in different buildings increases my stress | 172 (13.9) | 255 (20.5) | 245 (19.7) | 345 (27.8) | 224 (18.0) |
| On-campus exam increases my stress | 273 (22.0) | 303 (24.4) | 254 (20.5) | 255 (20.5) | 156 (12.5) |
| Presence of an exam observer increases my stress | 220 (17.7) | 245 (19.7) | 298 (24.0) | 311 (25.1) | 167 (13.5) |
| The higher study load during face-to-face learning increases my stress | 359 (28.9) | 403 (32.5) | 239 (19.3) | 156 (12.6) | 84 (6.7) |
| Having to focus with the lecturer for a long time increases my stress | 320 (25.8) | 384 (30.9) | 260 (21.0) | 204 (16.4) | 73 (5.9) |
| Not being able to attend the lectures at a time that suits me increases my stress | 339 (27.3) | 447 (36.0) | 251 (20.2) | 146 (11.8) | 58 (4.7) |
| **Online Learning** | | | | | |
| Technical issues increase my stress | 286 (23.0) | 468 (37.7) | 229 (18.5) | 193 (15.6) | 65 (5.2) |
| Dealing with new platforms increases my stress | 144 (11.6) | 304 (24.5) | 307 (24.7) | 343 (27.6) | 143 (11.5) |
| Exams difficulty increases my stress | 346 (27.9) | 398 (32.1) | 238 (19.2) | 192 (15.5) | 67 (5.4) |
| Unfairness in exam marks increases my stress | 463 (37.3) | 391 (31.5) | 225 (18.1) | 113 (9.1) | 49 (3.9) |
| Short exam time increases my stress | 491 (39.6) | 420 (33.8) | 197 (15.9) | 94 (7.6) | 39 (3.1) |
| Using 'one-way questions' during exams increases my stress | 741 (59.7) | 309 (24.9) | 131 (10.6) | 38 (3.1) | 22 (1.8) |
| Attending online lectures without my friends and lecturer increases my stress | 188 (15.1) | 271 (21.8) | 329 (26.5) | 303 (24.4) | 150 (12.1) |
| Not being able to directly discuss the educational material presented during lectures increases my stress | 239 (19.3) | 375 (30.2) | 309 (24.9) | 220 (17.7) | 98 (7.9) |
| Having a camera directed at me during the exam increases my stress | 472 (38.0) | 340 (27.4) | 248 (20.0) | 128 (10.3) | 53 (4.3) |

**Table 4.** Assessment of participants' stress ($n$ = 1241) via the Perceived Stress Scale (PSS).

| Statement | Never n (%) | Almost Never n (%) | Sometimes n (%) | Fairly Often n (%) | Very Often n (%) |
|---|---|---|---|---|---|
| In the last month, how often have you been upset because of something that happened unexpectedly? | 105 (8.5) | 176 (14.2) | 516 (41.6) | 336 (27.1) | 108 (8.7) |
| In the last month, how often have you felt that you were unable to control the important things in your life? | 113 (9.1) | 225 (18.1) | 410 (33.0) | 312 (25.1) | 181 (14.6) |
| In the last month, how often have you felt nervous and stressed? | 65 (5.2) | 198 (16.0) | 401 (32.3) | 334 (26.9) | 243 (19.6) |
| In the last month, how often have you felt confident about your ability to handle your personal problems? | 56 (4.5) | 208 (16.8) | 434 (35.0) | 335 (27.0) | 208 (16.8) |
| In the last month, how often have you felt that things were going your way? | 120 (9.7) | 330 (26.6) | 489 (39.4) | 218 (17.6) | 84 (6.8) |
| In the last month, how often have you found that you could not cope with all the things that you had to do? | 92 (7.4) | 263 (21.2) | 491 (39.6) | 264 (21.3) | 131 (10.6) |
| In the last month, how often have you been able to control irritations in your life? | 70 (5.6) | 278 (22.4) | 498 (40.1) | 262 (21.1) | 133 (10.7) |
| In the last month, how often have you felt that you were on top of things? | 96 (7.7) | 253 (20.4) | 505 (40.7) | 258 (20.8) | 129 (10.4) |
| In the last month, how often have you been angered because of things that were outside of your control? | 66 (5.3) | 232 (18.7) | 402 (32.4) | 331 (26.7) | 210 (16.9) |
| In the last month, how often have you felt difficulties were piling up so high that you could not overcome them? | 128 (10.3) | 245 (19.7) | 399 (32.2) | 269 (21.7) | 200 (16.1) |

Items 1, 2, 3, 6, 9, and 10 were coded as: 4: Never, 3: Almost never, 2: Sometimes, 1: Fairly often, and 0: Very often. Items 4, 5, 7, and 8 were coded as: 0: Never, 1: Almost never, 2: Sometimes, 3: Fairly often, and 4: Very often.

**Table 5.** Exploring factors affecting anger management among study participants (*n* = 1241).

| Parameter | Anger Management Issues | | | |
|---|---|---|---|---|
| | **Beta** | ***p*-Value** [#] | **Beta** | ***p*-Value** [$] |
| Age | −0.011 | 0.217 [^] | −0.018 | 0.050 * |
| **Gender** | | | | |
| • Male | Reference | | | |
| • Female | 0.187 | 0.005 [^] | 0.289 | <0.001 * |
| **Living place** | | | | |
| • Amman (the capital) | Reference | | | |
| • Other cities | 0.006 | 0.944 | — | — |
| **Living condition** | | | | |
| • Urban areas | Reference | | | |
| • Rural areas | 0.172 | 0.144 [^] | 0.190 | 0.099 |
| **Marital status** | | | | |
| • Married | Reference | | | |
| • Other (single, divorced, widowed) | <0.001 | 0.998 | — | — |
| **Smoking status** | | | | |
| • Non-smoker | Reference | | | |
| • Smoker | −0.344 | <0.001 [^] | −0.418 | <0.001 * |
| **Chronic Disease** | | | | |
| • No | Reference | | | |
| • Yes | 0.197 | <0.172 [^] | 0.063 | 0.696 |
| **Taking medication** | | | | |
| • No | Reference | | | |
| • Yes | 0.329 | 0.003 [^] | 0.273 | 0.027 * |
| **Cups of coffee (with caffeine) per day** | | | | |
| • Less than two cups | Reference | | | |
| • Two cups or more | 0.261 | 0.001 [^] | 0.175 | 0.029 * |
| **Having a family member diagnosed with anxiety or depression** | | | | |
| • No | Reference | | | |
| • Yes | 0.196 | 0.024 [^] | 0.130 | 0.131 |
| **Type of university** | | | | |
| • Private university | Reference | | | |
| • Public university | −0.055 | 0.418 | — | — |
| **Academic year** | | | | |
| • 3rd year or less | Reference | | | |
| • 4th year or above | 0.007 | 0.920 | — | — |

[#] Using simple linear regression. [$] Using multiple linear regression. [^] Eligible for entry in multiple linear regression (significant at 0.25 significance level). * Significant at 0.05 significance level.

## 4. Discussion

This is the first study that unveils university students' perceptions regarding the educational process they received post COVID-19 pandemic, and its impact on their psychological status. In this research work, 1241 students from four different Jordanian universities shared their perceptions regarding the educational process they went through and the stress level they perceived post-COVID-19. Most of the students who chose to participate were females,

living in Amman, single, non-smokers, had no chronic diseases, and were not using any medications. Results showed that the majority of the students, regardless of their academic background, preferred face-to-face over blended and online learning. Although most students declared that face-to-face learning methods were more efficient when compared to other learning methods, many of them reported that online learning is less stressful when compared to face-to-face or blended learning.

The theoretical framework used in developing and executing the surveys that involve the online collection of data from participants need to be carefully evaluated from many perspectives. The published study by Arco Bravo et al. involved the development of an online questionnaire that included 22 questions organized into three different sections including teaching strategies utilized, assessment, and teaching workload an emotional factor; the questionnaire was completed by 125 professors. Beyond simply transferring face-to-face teaching models to virtual ones, the authors conclude that it is necessary to implement formative proposals that help manage online or hybrid teaching [16]. Ramos-Pla A. et al. used a mixed methodology, involving a questionnaire with Likert-type, multiple, alternative, and open-ended responses. According to the study findings, collaborative work was the most commonly used teaching activity during the transition from face-to-face to online learning, followed by lectures via video conferences. The authors' advice for taking advantage of the pandemic's push toward virtual teaching is to continue opting for new technologies in higher education in order to face future emergency situations with resilience [17]. del Arco I et al. conducted a study involving an online questionnaire directed to students from 20 universities in Spain, with a total participation of 893 individuals. The study looked into students' perspectives on the relationship between the quality of learning during the pandemic period and the teaching plans, material resources, interaction processes, and affective-emotional component; the study concluded that universities must implement support and tutoring models, particularly for students in their first years at university, in order to develop competencies such as autonomy, digital competence, and self-regulation [18]. In the current study, an online survey consisting of four main sections with a range of 6–18 items in each section, whereas 28 of the total 50 items investigated stress-related factors; the study opened a new window overlooking the past and new educational methodologies and its association with stress levels amongst students. The new online educational methodology was found to cause less stress, a promising finding to the change brought about by the COVID-19 pandemic.

Being the educational methods of the future, face-to-face and online learning have been compared previously in many countries. According to a study conducted in Pakistan in 2020, students and faculty staff members reported that online learning was flexible and efficient, yet it had some limitations affecting the learning psychomotor skills of students, in addition to restricting their clinical and laboratory-gained skills [19]. A study conducted by Muhammad Adnan et al. in 2020 indicated that students perceived online learning to be a good alternative to face-to-face learning during the COVID-19 pandemic, but it was not as effective as face-to-face learning [20]. Students' perception towards the quality of online learning can be associated with the availability of the course materials, in that the more available the material was the better the quality of the online course rated [18]. Another study conducted by Ching-Hong Liu et al. in 2010 indicated that students who were enrolled in digital learning were more satisfied with the learning material and the learning environment than students receiving face-to-face education [21]. A meta-analysis conducted by Christian Ebner et al. in 2019 indicated that students' satisfaction towards face-to-face learning was superior to learning through webinars (a form of online learning); in addition, the satisfaction of students with online webinars was superior to their satisfaction with asynchronous online learning [22]. The results of this study support such previous findings, as students showed a preference for face-to-face learning over the online learning method. Although students believed that face-to-face learning was more efficient than online learning, a study conducted by Robert L. Smith et al. indicated that there is no difference in the level of education when comparing online

learning and face-to-face learning. Moreover, online learning was shown to have superior efficacy over face-to-face learning in certain cases, which is why online learning can be encouraged over face-to-face learning in certain theory-based study courses [23]. With this said, online learning can be associated with many challenges, including the inequality of accessibility to technological devices and a good internet connection, which can be affected by the socioeconomic status of the students and their families. The requirement of basic knowledge in technological skills, and the extra workload added to the academic staff in universities, add additional challenges. With this said, there were some activities implemented to cushion the impact of the pandemic on tertiary education, including the technological support some universities provided, and the research innovations that were implemented in response to the pandemic [24]. In Jordan, the assessment of students was also found to present an issue to many academics and students alike, which is consistent with other studies conducted in Jordan during the pandemic, all highlighting the lack of technological and assessment tools, and the absence of interactions during exams [25,26]. During the COVID-19 pandemic, a study conducted by Roberto Sánchez-Cabrero et al. included 919 master students in Spain and measured the difference between the assessment of students on-site and online, concluding that online assessment was legitimate and as reliable as the on-site assessment method. Moreover, the online assessment method was found to be more beneficial in the cases of online masters degrees [27]. When comparing the advantages and disadvantages of online learning during the COVID-19 pandemic, a study conducted in Turkey and Jordan assessed the pros and cons of online learning from a students' perspective and revealed that students believe online learning is more flexible and easily accessible, but comes with consequences such as technical problems related to the internet connection [28].

Stress affects all walks of life for students, and can affect the level of their learning and achievement. When it comes to perceived stress associated with learning, a study conducted in a liberal art college in the United States indicated that students undertaking face-to-face learning methods were significantly more stressed than their colleagues who were undertaking an online learning method [29]. Those results were consistent with the results of this study showing that students in Jordan felt less stress when learning via an online method compared to when receiving classes via the face-to-face learning method. In addition, this study indicated that being a female, living in rural areas, being a non-smoker, having a chronic disease, reporting a higher coffee intake, and having a family history of anxiety and depression, were all related to higher issues in anger management and thus higher stress levels.

Face-to-face learning may influence stressful events, such as the pressure to make new friends and the potential to feel isolated [30]. Online learning can be associated with stress due to other reasons; for instance, continuous exposure to smart devices is associated with burnout and stress, in addition to the physical burden of neck and back pain due to the long hours of using the computer or iPad [31]. In this study, students reported the obligation to attend classes on time and the high study load as the major sources of stress involved with face-to-face learning. As for online learning, examination procedures and the relatively short allocated examination time were reported as the most influential factors on students' stress levels. In general, the PSS score indicated that 62.9% of participants had moderate stress, while 22.4% had high stress, and 14.7% had low stress. The average PPS score was 20.88, which is relatively higher than the average score conveyed for university students in France (the average stress score reported was 15.9) [32], similar to that reported for university students in Nigeria (the average score was 20.76) [33], and lower than that stated for students in Malaysia (the average perceived score was 27.5) [34].

Converting to online learning has also increased stress and workload on university professors, a study conducted by Isabel del Arco Bravo et al. indicated that this conversion indeed increased the workload, in addition, to an increase in the emotional load among postgraduate professors [16]. From another perspective, the conversion to online learning

required more technical skills which resulted in an increase in courses taught by the professors according to a study conducted by Anabel Ramos-Pla et al. [35].

Researchers have been studying students' preferences regarding the method of learning and a constant question has always been asked about whether the future will be for online or face-to-face learning [36–38]. Forecasting which of the educational methods will be used most in the future based on the experience of students during and post-COVID-19 is enlightening and would create opportunities to implement more creative ways of learning. A study conducted by Anabel Ramos-Pla et al. concluded that online learning helped in engaging students in a more creative way [17]. In this study, most students believed that the future will involve a blended learning strategy of 50% online learning and 50% face-to-face learning. Previous research explored the influence of personality factors on the choice preference of online versus face-to-face learning. Students who recently experienced a positive online learning experience were found to prefer online learning courses [39], hence it was all based on students' past experience. Looking at the results of this study and other previous studies, it can be stated that online learning may be favored by students when compared to face-to-face learning in the future if they have had a previous positive experience [37,40]; other meta-analyses suggest that the difference in students' preference between the two educational methods is subtle [41].

This study has a limitation in that the impact of synchronous learning—in which instructors and students gather simultaneously in a physical location or virtual one to interact in "real-time"—and asynchronous learning—in which students access materials at their own pace and interact with one another over long periods of time—was not thoroughly assessed.

## 5. Conclusions

Although Jordanian students prefer face-to-face learning over online learning, they believe that online learning can be less stressful than face-to-face learning, and can be the preferred method of education in the future if a positive current experience is provided. Face-to-face learning presents a more stressful environment due to the high study load and the more stringent approach to attending lectures, while the stress of online learning is caused mainly by the continuous exposure to smart devices and the associated burnouts and physical burden of neck and back pain due to the long hours of sitting at a computer. The perceived stress scale score revealed that 63% of students experienced moderate stress and 22% experienced high stress. When it comes to the future, the majority of students believed that education will be blended to include both online learning and face-to-face learning, but online learning may dominate if current positive experiences are provided by the educational institution. Further studies are needed to unveil suitable strategies to upgrade online the learning experience and decrease stress associated with both online and face-to-face learning.

**Author Contributions:** Conceptualization, I.A.B., T.Y.A., N.M.O., F.A.-a.A.-h. and M.R.; methodology, I.A.B., T.Y.A., N.M.O., F.A.-a.A.-h. and M.R.; validation, I.A.B., T.Y.A., N.M.O., F.A.-a.A.-h. and M.R.; formal analysis, I.A.B.; investigation, I.A.B., T.Y.A., N.M.O., F.A.-a.A.-h. and M.R.; data curation, I.A.B., T.Y.A., N.M.O., F.A.-a.A.-h. and M.R.; writing—original draft preparation, I.A.B. and T.Y.A.; writing—review and editing, I.A.B., T.Y.A., N.M.O., F.A.-a.A.-h. and M.R.; visualization, I.A.B., T.Y.A., N.M.O., F.A.-a.A.-h. and M.R.; supervision, I.A.B., T.Y.A., N.M.O., F.A.-a.A.-h. and M.R.; project administration, I.A.B. and T.Y.A.; funding acquisition, I.A.B., T.Y.A., N.M.O., F.A.-a.A.-h. and M.R. All authors have read and agreed to the published version of the manuscript.

**Funding:** This research was funded by the [Deanship of Research and Higher Studies of the involved universities], grant number [DRGS-2021-2022-16], and the APC was funded by [Applied Science Private University (ASU), the University of Jordan (UJ), the Hashemite University (HU), and Princess Sumaya University for Technology (PSUT)].

**Institutional Review Board Statement:** The study was conducted in accordance with the Declaration of Helsinki, and approved by the Institutional Review Board of Applied Science Private University (protocol code: 2022-PHA-15; date of approval: 1 August 2022).

**Informed Consent Statement:** Informed consent was obtained from all subjects involved in the study. Written informed consent has been obtained from the participants to publish this paper.

**Data Availability Statement:** Data available on request due to privacy or ethical restrictions.

**Conflicts of Interest:** The authors declare no conflict of interest.

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
