# Peer review of "Assessing Learning Preferences of Tertiary Education Students in Jordan Post COVID-19 Pandemic and the Associated Perceived Stress"

_education, doi:10.3390/educsci12110829_

Round 1
Reviewer 1 Report
Dear authors,
First of all, I would like to congratulate you on your work, as there are few studies focusing on Jordan related to this topic and with such a large sample. However, I would like to make some comments in order to improve your manuscript:
- The title does not correspond to the objective/variables of the study. the title talks about Assessment of the impact of the COVID-19 pandemic on the educational process and in the objective To assess the learning preferences of students.
- There is a lack of many updated bibliographical references. Studies similar to yours have been carried out but in other contexts. It would be necessary to expand the bibliography of the theoretical framework. Therefore, I recommend you:
Bravo, I.d.A.; Flores-Alarcia, Ò.; González-Rubio, J.; Araneda, D.S.; Olivos, C.L. Workloads and Emotional Factors Derived from the Transition towards Online and/or Hybrid Teaching among Postgraduate Professors: Review of the Lessons Learned. Educ. Sci. 2022, 12, 666. https://doi.org/10.3390/educsci12100666
Ramos-Pla, A.; Reese, L.; Arce, C.; Balladares, J.; Fiallos, B. Teaching Online: Lessons Learned about Methodological Strategies in Postgraduate Studies. Educ. Sci. 2022, 12, 688. https://doi.org/10.3390/educsci12100688
Ramos-Pla, A.; del Arco, I.; Flores Alarcia, Ò. University Professor Training in Times of COVID-19: Analysis of Training Programs and Perception of Impact on Teaching Practices. Educ. Sci. 2021, 11, 684. https://doi.org/10.3390/educsci11110684
del Arco, I.; Flores, Ò.; Ramos-Pla, A. Structural Model to Determine the Factors That Affect the Quality of Emergency Teaching, According to the Perception of the Student of the First University Courses. Sustainability 2021, 13, 2945. https://doi.org/10.3390/su13052945
- It is not clear to me whether the Google form they used for the survey is reliable in relation to the data protection of the informants. This should be clarified.
- Tables 1 and 2 could be accompanied by graphs to make them easier to read.
- In the discussion, the same problems can be detected as in the theoretical framework. You should discuss your results with more current references and with studies similar to yours.
- You should check the bibliographical references. Some of them are not correctly referenced.
Author Response
Revisions for the Manuscript ID [education-2008511] entitled "Assessing the impact of COVID-19 pandemic on the educational process and the perceived stress of students in Jordan." submitted to Education Sciences Journal.
We would like to thank you for your valued comments.
We have revised the manuscript according to the comments and suggestions provided. Please find below our point-by-point responses to the comments.
Comments and Suggestions for Authors
REVIEWER 1
Dear authors,
First of all, I would like to congratulate you on your work, as there are few studies focusing on Jordan related to this topic and with such a large sample. However, I would like to make some comments in order to improve your manuscript:
C1. The title does not correspond to the objective/variables of the study. the title talks about Assessment of the impact of the COVID-19 pandemic on the educational process and in the objective To assess the learning preferences of students.
R1. The title was changed as follows: “Assessing learning preferences of tertiary education students in Jordan post COVID-19 pandemic and the associated perceived stress.”
C2. There is a lack of many updated bibliographical references. Studies similar to yours have been carried out but in other contexts. It would be necessary to expand the bibliography of the theoretical framework. Therefore, I recommend you:
Bravo, I.d.A.; Flores-Alarcia, Ò.; González-Rubio, J.; Araneda, D.S.; Olivos, C.L. Workloads and Emotional Factors Derived from the Transition towards Online and/or Hybrid Teaching among Postgraduate Professors: Review of the Lessons Learned. Educ. Sci. 2022, 12, 666. https://doi.org/10.3390/educsci12100666
Ramos-Pla, A.; Reese, L.; Arce, C.; Balladares, J.; Fiallos, B. Teaching Online: Lessons Learned about Methodological Strategies in Postgraduate Studies. Educ. Sci. 2022, 12, 688. https://doi.org/10.3390/educsci12100688
Ramos-Pla, A.; del Arco, I.; Flores Alarcia, Ò. University Professor Training in Times of COVID-19: Analysis of Training Programs and Perception of Impact on Teaching Practices. Educ. Sci. 2021, 11, 684. https://doi.org/10.3390/educsci11110684
del Arco, I.; Flores, Ò.; Ramos-Pla, A. Structural Model to Determine the Factors That Affect the Quality of Emergency Teaching, According to the Perception of the Student of the First University Courses. Sustainability 2021, 13, 2945. https://doi.org/10.3390/su13052945
R2. The above references were added to the discussion to expand the bibliography of the framework of the paper.
C3. It is not clear to me whether the Google form they used for the survey is reliable in relation to the data protection of the informants. This should be clarified.
R3. The following sentence was added: “This cross-sectional study was conducted via a survey presented to the potential participant via a web-based Google forms. The web-based Google Forms used is considered to be a reliable platform with regard to data protection as only the main researcher who has access to the collected data from the participants can enter the participants’ responses. Moreover, this was clarified to the students prior to their participation through the informed consent paragraph added at the beginning of the survey”
C4. Tables 1 and 2 could be accompanied by graphs to make them easier to read.
R4. Done. Figure 1 was created.
C5. In the discussion, the same problems can be detected as in the theoretical framework. You should discuss your results with more current references and with studies similar to yours.
R5. The following paragraph was added to the discussion to highlight differences and similarities in the theoretical framework covered in the previous recommended studies and the current study: “The theoretical framework used in developing and executing the surveys that involve the online collection of data from participants need to be carefully evaluated from many perspectives. The published study by Arco Bravo et al. involved the development of an online questionnaire that included 22 questions organized into three different sections including teaching strategies utilized, assessment, and teaching workload an emotional factor; the questionnaire was completed by 125 professors. Beyond simply transferring face-to-face teaching models to virtual ones, the authors conclude that it is necessary to implement formative proposals that help manage online or hybrid teaching (16). Ramos-Pla A. et al used a mixed methodology, involving a questionnaire with Likert-type, multiple, alternative, and open-ended responses. According to the study findings, collaborative work was the most commonly used teaching activity during the transition from face-to-face to online learning, followed by lectures via videoconferences. The authors’ advice for taking advantage of the pandemic's push toward virtual teaching is to continue opting for new technologies in higher education in order to face future emergency situations with resilience (17). del Arco I et al conducted a study involving an online questionnaire directed to students from 20 universities in Spain, with a total participation of 893 individuals. The study looked into students' perspectives on the relationship between the quality of learning during the pandemic period and the teaching plans, material resources, interaction processes, and affective-emotional component; The study concluded that universities must implement support and tutoring models, particularly for students in their first years at university, in order to develop competencies such as autonomy, digital competence, and self-regulation (18). In the current study, an online survey consisting of four main sections with a range of 6-18 items in each section, whereas 28 of the total 50 items investigated stress-related factors; the study opened a new window overlooking the past and new educational methodologies and its association with stress levels amongst students. The new online educational methodology was found to cause less stress, a promising finding to the change brought about by the COVID-19 pandemic.”
C6. You should check the bibliographical references. Some of them are not correctly referenced.
R6. All the bibliographical references were checked and corrected in the case an error was detected
Reviewer 2 Report
nice research. Your study deals with online education, did you differentiate between synchronous and asynchronous? Would that have impacted results?
Author Response
Revisions for the Manuscript ID [education-2008511] entitled "Assessing the impact of COVID-19 pandemic on the educational process and the perceived stress of students in Jordan." submitted to Education Sciences Journal.
We would like to thank you for your valuable comments.
We have revised the manuscript according to the comments and suggestions provided. Please find below our point-by-point responses to the comments.
REVIEWER 2
C1. nice research. Your study deals with online education, did you differentiate between synchronous and asynchronous? Would that have impacted results?
R1. Thank you, we agree with your comment. The following point was added as a limitation of the study, as follows: “This study has a limitation in that the impact of synchronous learning—in which instructors and students gather simultaneously in a physical location or virtual one to interact in "real-time"—and asynchronous learning—in which students access materials at their own pace and interact with one another over longer periods of time—was not thoroughly assessed.”